# The Development of a Series of Genomic DNA Reference Materials with Specific Copy Number Ratios for The Detection of Genetically Modified Maize DBN9936

**DOI:** 10.3390/foods13050747

**Published:** 2024-02-28

**Authors:** Jun Li, Hongfei Gao, Yunjing Li, Shanshan Zhai, Fang Xiao, Gang Wu, Yuhua Wu

**Affiliations:** Key Laboratory of Agricultural Genetically Modified Organisms Traceability of the Ministry of Agriculture and Rural Affairs, Oil Crops Research Institute, Chinese Academy of Agricultural Sciences, Wuhan 430062, China; lijuner@caas.cn (J.L.); gaohf7@126.com (H.G.); liyunjing@caas.cn (Y.L.); zhaishanshan@caas.cn (S.Z.); xiaofang@caas.cn (F.X.); wugang@caas.cn (G.W.)

**Keywords:** DBN9936, genomic DNA, certified reference material, duplex droplet digital PCR

## Abstract

The genetically modified (GM) maize DBN9936 with a biosafety certificate will soon undergo commercial application. To monitor the safety of DBN9936 maize, three genomic DNA (gDNA) reference materials (RMs) (DBN9936a, DBN9936b, and DBN9936c) were prepared with nominal copy number ratios of 100%, 3%, and 1% for the DBN9936 event, respectively. DBN9936a was prepared from the leaf tissue gDNA of DBN9936 homozygotes, while DBN9936b and DBN9936c were prepared by the quantitative mixing of gDNA from the leaf tissues of DBN9936 homozygotes and non-GM counterparts. Validated DBN9936/*zSSIIb* duplex droplet digital PCR was demonstrated to be an accurate reference method for conducting homogeneity study, stability study, and collaborative characterization. The minimum intake for one measurement was determined to be 2 μL, and the gDNA RMs were stable during transport at 37 °C for 14 days and storage at −20 °C for 18 months. Each gDNA RM was certified for three property values: DBN9936 event copy number concentration, *zSSIIb* reference gene copy number concentration, and DBN9936/*zSSIIb* copy number ratio. The measurement uncertainty of the certified values took the uncertainty components related to possible inhomogeneity, instability, and characterization into account. This batch of gDNA RMs can be used for calibration and quality control when quantifying DBN9936 events.

## 1. Introduction

Since the first genetically modified organisms (GMOs) were approved for commercialization, there has been increasing concern among consumers about the potential risks associated with consuming GMOs. Many countries have introduced legislation to regulate the labeling of GMO-containing products; however, the threshold levels for the labeling of products containing GMOs differ from country to country. For example, the European Union (EU) requires all food and feed products that contain more than 0.9% of authorized genetically modified (GM) crops to be labeled [1], whereas the threshold value is 5% in Japan and Canada, 3% in South Korea, and 1% in Australia and New Zealand [2]. China will soon stipulate a threshold of 3% for the labeling of GMO-containing products to promote the industrialization of biological breeding.

Implementing a policy for the labeling of GMO-containing products is crucial for ensuring the safety and transparency of the food and feed supply chains. So, too, is establishing a GMO quantification process that uses standardized analytical methods and certified reference materials (CRMs) at certified values. In addition, during GMO quantification, an appropriate CRM should be used to ensure the analytical traceability of the measurement results and to verify the performance of the measurement process [3].

CRMs designated for DNA-based analytical methods are reference materials (RMs) with a certificate, and they can be powdered materials that contain the analyte, genomic DNA (gDNA) extracted from material that contains the analyte, or plasmids that contain the nucleotide sequence of the specific analyte [4]. The development of GMO CRMs requires a global effort, and various organizations and agencies are working to ensure the accuracy and reliability of GMO quantification results. The EU has been actively involved in the development of GMO CRMs since the early 2000s, and the GMO CRMs produced by the EU mainly include matrix CRMs and plasmid CRMs (https://crm.jrc.ec.europa.eu/, accessed on 7 March 2023). Matrix CRMs are usually composed of a series of CRMs with nominal mass fractions of the GMO, such as 0, 0.1, 1, 10, and 100% (*m*/*m*) [5]. However, only CRMs with 0 and 100% GMO mass fractions are available for GM potato and sugarcane [6,7]. By 2023, the EU had released 153 types of matrix CRMs for GMO detection that cover 37 GM events approved by the EU, as well as four plasmid CRMs for the detection of GM soybean 356043 and GM corn MON810, NK603, and 98140 [8]. In the USA, GMO CRMs available as pure powder or pure leaf tissue gDNA are developed by GMO developers and released by the American Oil Chemists’ Society (AOCS) (https://www.aocs.org/crm?SSO=True, accessed on 7 March 2023). The GMO CRMs released by the AOCS are assigned according to purity, and the uncertainty associated with the quantity value is ignored. To date, 73 CRMs have been released by the AOCS, including 21 gDNA RMs and 52 powdery RMs, covering seven crops: GM rape, cotton, corn, potato, rice, soybean, and sugar beet. In addition, the company Nippon Gene produces some plasmid RMs that are designed to be used as controls [9]. To meet the requirements set for the labeling of GMO-containing products, China has developed 47 GMO CRMs: 34 matrix CRMs, 5 gDNA CRMs, and 8 plasmid CRMs (https://www.ncrm.org.cn/Web/Material/List?fenleiAutoID=14&pageIndex=1, accessed on 7 March 2023).

CRMs comprising gDNA have the advantages of requiring a small amount of raw materials for their production, simple preparation, and convenient application. The gDNA RMs currently available are all pure RMs prepared from GM homozygous plants [10,11], which means that they meet the requirements for use as calibrators in quantitative real-time PCR (qPCR) assays. In addition to the need for calibration (and calibrators) in the quantification process, there is also a need for quality control to determine whether there is any bias in the measurement, and the CRM used for calibration should not also be used for bias control [3,12,13]. Thus, it is essential to develop low-level gDNA CRMs equivalent to threshold values as well as pure gDNA CRMs to meet the requirements of both calibration and bias control in GMO quantification.

The GM maize DBN9936 is an insect-resistant and herbicide-tolerant variety that was developed by the Beijing DaBeiNong Biotechnology Company in China using an *Agrobacterium*-mediated method, and it harbors a *CP4-EPSPS* and a *Cry1Ab* expression cassette. This GM maize variety was granted biosafety certification in China in 2019 and was approved for food safety prior to market entry by the US Food and Drug Administration in 2021. A successful industrialization pilot was performed with DBN9936 maize in China, and it will soon be planted commercially and used for food and feed. The surveillance of the DBN9936 event requires the development of corresponding CRMs and a standardized quantification method. Therefore, the aim of this study was to develop a series of gDNA CRMs with specific copy number ratios that can be used for calibration and quality control during DBN9936 maize detection and quantification. Through the production of these DBN9936 maize gDNA CRMs, we also aimed to establish a platform for producing a series of GMO gDNA CRMs with graduated copy number ratios.

## 2. Materials and Methods

### 2.1. Raw Materials

DBN9936 and recipient DBN318 seeds were provided by DabeiNong Biotechnology Co., Ltd. (Beijing, China). After confirming that the DBN9936 event was present in the GM seeds and absent from the non-GM seeds, both the GM and non-GM seeds were sown in culture pots and cultured in a greenhouse. When seedlings had grown, leaves were collected from individual plants for subsequent testing.

### 2.2. Zygosity Testing of Genetically Modified Plants

It was necessary to identify the GM homozygotes, as they were to be used as the raw material in the preparation of the gDNA RMs. The gDNA of the seedlings was subjected to zygosity testing using real-time PCR. The DBN9936 event-specific primer/probe set LF51/LR126/LP79 was used to test for the presence of the DBN9936 event-specific sequence, and the primer/probe set IF/IR/IP was used to test for the absence of an exogenous insert sequence at the insertion site of the recipient genome (Figure 1). The primer and probe sequences are shown in Appendix A. The GM homozygous, GM heterozygous, and non-GM plants were distinguished by the resultant fluorescence curves. Material from GM homozygotes was only amplified in reactions that contained the LF51/LR126/LP79 primer/probe set, material from heterozygotes was both amplified in reactions that contained either of two primer/probe sets, and material from non-GM plants was only amplified in reactions that contained the IF/IR/IP primer/probe set (Figure 1). The GM homozygotes were labeled and allowed to continue growing.

### 2.3. Large-Scale Extraction of gDNA

During the flourishing period of the plants’ vegetative growth, leaves were collected for large-scale extraction of gDNA. The collected leaves were frozen in liquid nitrogen or in a −70 °C freezer. To avoid cross contamination, the GM and non-GM materials were milled separately into powder using a Freezer Mill Spex6870 (SPEX SamplePrep; Thermo Fischer Scientific, Metuchen, NJ, USA). The samples placed in the grinding vial were pulverized with a magnetically driven impactor, and the grinding vial was kept in liquid nitrogen throughout the process. After the leaves were milled to the required particle size, the powder was transferred into 50 mL centrifuge tubes for DNA extraction. An improved hexadecyltrimethyl ammonium bromide (CTAB) method was used to extract the gDNA [11]. The extracted gDNA was dissolved in 0.1 × TE. The integrity of the gDNA was determined using agarose gel electrophoresis, and the quality of the gDNA was evaluated using the OD260/280 and OD230/260 values calculated from UV absorbance measurements (NanoDrop 2000; Thermo Fischer Scientific, Waltham, MA, USA).

### 2.4. Preparation of the gDNA RMs

The copy number concentrations of the GM and non-GM gDNA were measured accurately via *zSSIIb* droplet digital PCR (ddPCR), with at least triplicate measurements. The concentration of the GM gDNA was adjusted to be identical to that of the non-GM gDNA, approximately 100 ng/μL. The extracted gDNA from the GM homozygotes was used directly to prepare the pure gDNA RM termed DBN9936a. The GM and non-GM gDNA samples were weighed on a calibrated balance and then mixed to prepare the DBN9936b and DBN9936c RMs, which had nominal copy number ratios of 3% and 1% of the DBN9936 event, approximately equivalent to the threshold values of China and the EU, respectively. The resultant homogenized gDNA solutions were packaged into sterile 2 mL skirted screw-cap tubes in aliquots of 100 μL in biological safety cabinets. The tubes were sealed and stored at −70 °C.

### 2.5. Homogeneity Study

In accordance with ISO Guide 35:2017 [14], 15 units were randomly sampled from the entire batch of each gDNA RM for a between-unit homogeneity study; the sampling covered the initial, intermediate, and final stages of the RM packaging. Three sub-samples were taken from different positions in each unit for the homogeneity test. The DBN9936/*zSSIIb* duplex ddPCR method developed and validated in a previous study was used [15]. The duplex ddPCR was conducted on the QX200™ Droplet Digital™ PCR platform (Bio-Rad, Pleasanton, CA, USA) with 20 µL of the initial PCR mixture containing 1 × ddPCR Supermix (No dUTP) (Bio-Rad), 400 nM primers, and 200 nM probes, using the following program: 95 °C for 10 min, 50 cycles of 94 °C for 30 s, 57 °C for 60 s, and 98 °C for 10 min. The acquired data were statistically analyzed using an analysis of variance (ANOVA) in Microsoft Excel, and the mean square between units (*MS*_between_) and the mean squares within units (*MS*_within_) were calculated. The *MS*_between_ to *MS*_within_ ratio (*F* value) was then calculated and compared to the critical value of *F*_α_. Between-group homogeneity was demonstrated when *F* < *F*_α_.

When *MS*_between_ was greater than *MS*_within_, the measurement uncertainty (*u*_bb_) introduced by the between-unit inhomogeneity was estimated using Equation (1):(1)$$u_{bb}=s_{bb}=\sqrt{\frac{{MS}_{between}-{MS}_{within}}{n}}$$
where *n* represents the mean number of replicates analyzed per unit.

When *MS*_between_ was less than *MS*_within_, the maximum inhomogeneity may have been hidden by the intermediate precision, and the *u*_bb_ introduced by the between-unit inhomogeneity was estimated using Equation (2) [16]:(2)$$u_{bb}=\sqrt{\frac{{MS}_{within}}{n}}\sqrt[{4}]{\frac{2}{v_{{MS}_{within}}}}$$
where *n* represents the mean number of replicates analyzed per unit $v_{{MS}_{within}}$ and represents the degrees of freedom of ${MS}_{within}$.

### 2.6. Stability Assessment

Stability was assessed according to ISO Guide 35:2017 [14]. For the short-term stability study, the gDNA RMs were stored at 4, 25, and 37 °C for 0, 3, 7, and 14 days. For the long-term stability study, the gDNA RMs were stored at −20 °C for 0, 1, 2, 4, 6, 12, and 18 months. The reference temperature was set to −70 °C. Three units per storage time were randomly selected for measurement via DBN9936/*zSSIIb* duplex ddPCR. An isochronous design was adopted to perform the short-term stability study; at the end of the isochronous storage, the sampled units were analyzed simultaneously under intermediate precision conditions [17]. An isochronous design together with a classical design was adopted to perform the long-term stability study; the units sampled between 0 and 6 months were analyzed simultaneously under intermediate precision conditions, and the units stored for 12 and 18 months were analyzed separately after the storage period. The data acquired during the stability study were evaluated against the storage time for each temperature, and the regression lines for plots of property values versus time were calculated, as shown in Equation (3):(3)$$Y=\beta_{1}X+\beta_{0}$$
where Y = property value, = slope, = intercept, and *X* = time.

Student’s *t*-test was then performed to determine whether the slopes of the regression lines showed statistical significance at a 95% confidence level. The uncertainties associated with short-term stability (*u*_sts_) and long-term stability (*u*_lts_) were estimated as the product of the storage time (*t*) and the uncertainty of the regression lines (s(*β*_1_)), as shown in Equation (4) [18]:(4)$$u_{lts}\left(u_{sts}\right)=s(\beta_{1})\times t$$

For the freeze–thaw stability study, two units were sampled from the units stored at −70 °C and naturally thawed at room temperature. When the units had completely melted, 20 μL of DNA solution was taken from each unit and moved to −70 °C. The property values of the sub-samples after one to five freeze–thaw cycles were simultaneously measured under repeatability conditions. A *t*-test was performed to evaluate the variance of the property values with the freeze–thaw cycles.

### 2.7. Collaborative Characterization

At least eight qualified laboratories were invited to characterize the gDNA RMs using the DBN9936/*zSSIIb* duplex ddPCR method. Two units of each gDNA RM plus the primers/probes were mailed to each participant in dry ice, and the related ddPCR reagents were prepared by each participant. The participants were requested to measure the copy number concentrations of the DBN9936 event and *zSSIIb* reference gene, as well as the DBN9936/*zSSIIb* copy number ratio of the units using the specified protocol. Each unit was measured in at least quadruplicate ddPCR assays, and at least eight independent results for each property value were provided by each participant. According to ISO Guide 35:2017 [14], after measurement, the raw file from the ddPCR assay should be exported and sent to the organizer for statistical analysis. The Dixon test was performed to screen for intra-laboratory and inter-laboratory outliers, the D’Agostoon test was performed to determine whether the characterized data were normally distributed, and the Cochran test was performed to detect outlying standard deviation (SD) values.

The property values of the gDNA RMs were characterized by DBN9936/*zSSIIb* duplex ddPCR assay. The calculation of the copy number concentration of the DNA template was based on the copy number of the template per partition droplet volume (*v*), and dilution factor of the template (*d*) [19].

The uncertainty related to characterization (*u*_char_) mainly considered the contribution of the precision data (*u*_A_), partitioning of the sample (*u*_λ_), droplet volume variation (*u*_v_), and dilution factor (*u*_d_) [20]. The uncertainty of the arithmetic mean (*u*_A_) was estimated from the precision data evaluated by a statistical analysis of the collaborative measurements by the eight laboratories, the uncertainty related to the number of template molecules per droplet (*u*_λ_) was estimated according to the published method [21], the droplet volume uncertainty (*u*_v_) was directly estimated using data from published studies [22], and the dilution factor uncertainty (*u*_d_) (including the uncertainty related to the dilution factor of the DNA sample before adding to the PCR mix and of the DNA solution in the PCR mix) was estimated on the basis of the calibration uncertainty of the pipette. The uncertainty contributions were combined into the measurement uncertainty of the characterization by calculating the square root of the sum of squares of all components, as shown in Equation (5) [14]:(5)$$u_{char,r}=\sqrt{u_{char1,r}^{2}+u_{\lambda ,r}^{2}+u_{v,r}^{2}+u_{d,r}^{2}}$$

### 2.8. Estimation of Expanded Uncertainties

The combined uncertainty of the certified value (*Y*) consisted of uncertainty components from the characterization (*u*_char_), the potential between-unit inhomogeneity (*u*_bb_), and the potential instability during dispatch (*u*_sts_) and long-term storage (*u*_lts_). These four contributions were combined to estimate the expanded uncertainty of the certified value (*U*_CRM_) with coverage factor *k* (*k* = 2 at 95% confidence level) following Equation (6):(6)$$U_{CRM}=Y\cdot k\cdot \sqrt{u_{bb,r}^{2}+u_{sts,r}^{2}+u_{lts,r}^{2}+u_{char,r}^{2}}$$

## 3. Results

### 3.1. Genomic DNA Extraction and Quality Assessment

The zygosity of the GM plants determines the copy number ratio of the DBN9936 event to the *zSSIIb* reference gene (DBN9936/*zSSIIb*) in the gDNA of the raw materials. Theoretically, the DBN9936/*zSSIIb* copy number ratio in a GM homozygote is 1.0, equivalent to 100% GMO content, and that in a heterozygote is 0.5. The DBN9936 homozygotes and non-GM plants were individually identified by real-time PCR using the LF51/LR126/LP79 and IF/IR/IP primer/probe sets (Appendix A). Large-scale gDNA extraction was performed to collect gDNA from the leaves of the DBN9936 homozygotes and non-GM plants.

The integrity and purity of gDNA are closely related to the applicability and stability of gDNA RMs. The electrophoresis results showed that the gDNA from both the GM and non-GM plants exhibited good integrity, without obvious smears. The OD260/OD280 ratio values of the gDNA solutions were calculated from the absorbance measurements and found to be within the range of 1.8–1.9, and the OD260/OD230 ratio values were greater than 2.0. No obvious impurities of RNA, proteins, chaotropic salts, and phenol were detected in the gDNA solution. Since the extraction process is not robust enough to completely eliminate all impurities from the gDNA sample in practice, trace impurities may have been present in the DNA template that inhibited the PCR. Hence, real-time PCR assays designed to amplify and detect the DBN9936 event and *zSSIIb* were performed to assess the impact of impurities on the efficiency of the PCR using five serially diluted gDNA solutions as templates. The amplification efficiencies of the DBN9936 event and *zSSIIb* were calculated to be 98.14 and 101.8%, respectively, which were within the acceptable range of 90–110% [23]. The amplification efficiency assessment demonstrated that there were no inhibitors in the gDNA solution.

### 3.2. Preparation of gDNA RMs

The copy number concentrations of the DBN9936 gDNA and non-GM maize gDNA were accurately measured by *zSSIIb* ddPCR assay. The copy number concentration of the non-GM gDNA was adjusted to be close to that of the DBN9936 gDNA according to the measurement result. After adjustment, the copy number concentration of the DBN9936 gDNA was 33,712 copies/μL with an SD of 720 copies/μL (*n* = 8), and that of the non-GM gDNA was 33,812 copies/μL with an SD of 945 copies/μL (*n* = 8).

Three gDNA RMs with different copy number ratios were prepared for the DBN9936 event. The DBN9936 gDNA from the homozygote plants was used directly to prepare the pure gDNA RM termed DBN9936a. The gDNA from the DBN9936 maize plants and that from the non-GM recipients had the same genetic background, and their concentration was consistent. Therefore, it was concluded that the two gDNA solutions had the same density. The other two gDNA RMs—DBN9936b at 3% and DBN9936c at 1%—were prepared gravimetrically with the use of a calibrated mass balance and by taking into account the copy number concentrations of the two gDNA solutions.

DBN9936b and DBN9936c were prepared by quantitatively mixing the gDNA of DBN9936 and that of its non-GM counterpart; the mixture was thoroughly homogenized at 4 °C before packaging. Nine samples were taken from different positions in each gDNA mixture for an initial homogeneity assessment by DBN9936/*zSSIIb* duplex ddPCR assay after 24 h of mixing. The following three property values were measured for each sample: DBN9936 copy number concentration, *zSSIIb* copy number concentration, and DBN9936/*zSSIIb* copy number ratio. An *F*-test was performed with the measurement data, and the results showed that the statistical *F* value of each property was less than the critical value of 2.51 (*F*_(0.05,8,18)_) for the three gDNA RMs (Appendix A). The results of this initial homogeneity assessment showed that the gDNA mixtures had been thoroughly homogenized. DBN9936a was packaged into 500 vials, while DBN9936b and DBN9936c were each packaged into 300 vials.

### 3.3. Homogeneity Assessment

A key requirement for a batch of RMs is equivalence of property values among different units. Therefore, the standard (ISO 17034:2016) requires RM producers to assess the between-unit homogeneity to ensure that the certified values are valid within the stated uncertainties for all the units of a CRM [24]. The between-unit homogeneity was assessed for DBN9936a, DBN9936b, and DBN9936c based on the DBN9936 event copy number concentration, *zSSIIb* gene copy number concentration, and DBN9936/*zSSIIb* copy number ratio. The values of each parameter were quantified simultaneously by DBN9936/*zSSIIb* duplex ddPCR under repeatability conditions on units taken randomly from the entire batch and analyzed in a randomized manner. Grubbs’ test was used to identify outliers, and no outliers were identified in the individual results and unit means (at a confidence level of 95%).

The quantitative data obtained for the three property values of each RM were statistically analyzed by ANOVA (Appendix A). The *F* values were calculated by dividing the mean square between units (*MS*_between_) by the mean squares within units (*MS*_within_), and they were found to be less than the critical value of *F*
_0.05(14,30)_ at a 95% confidence level (Table 1). The ANOVA results indicated that the gDNA RMs had sufficient between-unit homogeneity across the three property values. The uncertainty introduced by between-unit inhomogeneity (*u*_bb_) is usually quantified as equivalent to the between-unit variation (*s*_bb_) (Equation (1)), which is separated from the within-unit variation (*s*_wb_) in the ANOVA. Both the between-unit standard deviations (*s*_bb_) and within-unit standard deviations (*s*_wb_) are subject to random fluctuations. When the *MS*_between_ was less than the *MS*_within_, or the calculated *s*_bb_ was less than *s*_wb_, the maximum inhomogeneity (*u*_bb_) could have been hidden by the method repeatability standard deviation (equivalent to the s_wb_), and Equation (2) was adopted to estimate the measurement uncertainty (*u*_bb_^*^) introduced by the between-unit inhomogeneity [16]. The larger values of *s*_bb_ and *u*_bb_^*^ were adopted as the uncertainty contribution (*u*_bb_) to account for potential inhomogeneity. The results of the evaluation of *s*_wb_, *s*_bb_, and *u*_bb_ are summarized in Table 1.

The minimum sample size that is representative of the whole unit is correlated to the within-unit homogeneity. To guarantee the certified value within its stated uncertainty, a sample equal to or above the minimum sample size should be used in quantitative PCR assays. When studying the between-unit homogeneity of this batch of RMs, 2 μL aliquots of gDNA were used in the PCRs. The PCR assay results confirmed that the precision was acceptable and that the between-unit homogeneity among this batch of RMs was sufficient. Therefore, the minimum sample intake for this batch of gDNA RMs was determined to be 2 μL.

### 3.4. Stability Assessment

It is necessary to conduct a stability test to establish the long-term storage conditions and short-term dispatch conditions of gDNA RMs. Moreover, repeated freezing and thawing could affect the stability of gDNA RMs. Typically, short-term, long-term, and freeze–thaw stability are examined in a stability assessment. The RMs assessed in this study consisted of gDNA solutions that were dispatched in ice boxes and stored in refrigerators; thus, the possibility of degradation due to light exposure was not a factor. Therefore, time and temperature were the two key factors that may have affected the stability of the RMs.

The short-term and long-term stability data were first screened for outliers using Grubbs’ test. Then, the data were evaluated against storage time for each temperature. For this, the slopes of the resultant regression lines for copy number concentration and copy number ratio versus storage time were calculated to determine whether there were any increases or decreases in the property values over time at different temperatures (Table 2 and Table 3). The slopes of the regression lines were assessed using the *t*-test for statistical significance. The calculated standard deviations of the slopes of the regression lines are shown in Table 2 for each RM at the different short-term dispatch temperatures and in Table 3 for each RM at the long-term storage temperature. The product of *S*(*β*_1_) and the *t*_0.95,n−2_ value was greater than *t*_0.95,2_ = 4.30 in the short-term stability study and *t*_0.95,5_ = 2.57 in the long-term stability study. The *t*-test results demonstrated that the slopes of the regression lines were not significantly different from zero, and none of the observed trends in the property values were statistically significant (at a 95% confidence level) for the three gDNA RMs at any of the tested temperatures. No significant degradation or volatilization of the gDNA was observed during the long-term storage or when samples were dispatched at 37 °C for 14 days. The findings indicated that the gDNA RMs could be transported for 14 days in ambient conditions below 37 °C and that adding ice or dry ice to the transportation box effectively ensured that the temperature remained below 37 °C during dispatch. The findings also showed that the gDNA RMs could be stored at −20 °C for at least 18 months.

In terms of freeze–thaw stability, previous studies with gDNA CRMs revealed that their property values remained steady after 10 freeze–thaw cycles [10,11]. When a gDNA RM is used to construct standard curves for the transgene and reference gene, at least 20 μL of gDNA solution is required. There was no gDNA sample volume left after a total of five freeze–thaw cycles for each 100 μL unit. The *t*-test results showed that the copy number concentration and copy number ratio values recorded after two, three, four, and five freeze–thaw cycles did not significantly differ from the values recorded after one freeze–thaw cycle for the three gDNA RMs (Appendix A). The findings suggest that a gDNA RM unit should be exhausted after five freeze–thaw cycles.

The occurrence of gDNA degradation during dispatch and storage cannot be eliminated entirely, although statistically significant trends were not observed. The uncertainties related to the stability of each property value during transportation (*u*_sts_) (Table 2) and storage (*u*_lts_) were estimated for the three gDNA RMs (Table 3). The uncertainty associated with short-term stability was estimated using the data collected when the RMs were kept at 37 °C for 14 days. The uncertainty associated with long-term stability was estimated using the data collected when the RMs were kept at −20 °C for 18 months. Hence, *u*_sts_ and *u*_lts_ represent the possible degradation that could occur during transport for 14 days at 37 °C and during storage for 18 months at −20 °C, respectively.

### 3.5. Characterization

Determining the DBN9936 event copy number concentration, *zSSIIb* copy number concentration, and DBN9936/*zSSIIb* copy number ratio values of DBN9936a, DBN9936b, and DBN9936c is an essential step in the development of DBN9936 maize CRMs. Given that earlier results showed that duplex ddPCR could be used to directly quantify the above-mentioned property values, it was selected as the characterization method to be used by the participating laboratories. Eight qualified laboratories were selected to participate in the collaborative characterization. After they performed the characterization, each laboratory forwarded their measurement data and the raw ddPCR files to the study organizer. Each laboratory provided at least eight independent measurement data for each property value for each of the three gDNA RMs, according to the study organizer’s requirement. A statistical analysis of the collaborative measurement data showed that there were no intra-laboratory outliers within the data from the eight participants, that there were no inter-laboratory mean or SD outliers, and that the measurement data from the eight laboratories displayed a normal distribution. The arithmetic means of the measurement data from the eight laboratories were assigned as the certified values of the three gDNA RMs (Appendix A). The certified DBN9936 event copy number concentration, *zSSIIb* copy number concentration, and DBN9936/*zSSIIb* copy number ratio values for DBN9936a, DBN9936b, and DBN9936c are summarized in Table 4.

Based on the precision data of the DBN9936 event and *zSSIIb* copy number concentrations of the three RMs and considering the mathematical model, the estimated measurement uncertainty for the collaborative characterization was established [25]. All the uncertainty components (i.e., *u*_A_, *u*_λ_, *u*_v_, and *u*_d_) associated with the characterization of the copy number concentration were considered and estimated (Table 4). The relative precision uncertainty (*u*_A,rel_), provided by the total variation in each RM, varied from 0.014 to 0.043 and correlated with the uncertainty provided by the λ factor. The uncertainty provided by the λ factor (*u*_λ_) was reported to be minimized at an optimal template concentration of approximately 1.59 molecules per partition [26,27]. The estimated *u*_λ,rel_ increased from 0.010 to 0.063 as the value of λ decreased from 1.577 to 0.015 (Table 4). DBN9936c had the lowest DBN9936 copy number concentration as well as the lowest λ value, and the related uncertainty was estimated to be a maximum of 0.063. Since the ddPCR assays were conducted under duplex conditions, the uncertainties associated with the droplet volume and dilution factor were not considered in the estimation of the uncertainty of the copy number ratio (*u*_char_), which only consisted of the uncertainty of the precision data evaluated by statistical analysis of a series of measurements (*u*_A_), and the uncertainties related to the *λ* values of the DBN9936 event and *zSSIIb* (*u*_λ_) [3]. All the uncertainty components were then combined to obtain the relative standard uncertainty related to the characterization of the DBN9936 event copy number concentration, *zSSIIb* copy number concentration, and DBN9936/*zSSIIb* ratio of the three RMs (Table 4).

### 3.6. Value Assignment

Each gDNA CRM had three certified values assigned for the DBN9936 event copy number concentration, *zSSIIb* copy number concentration, and DBN9936/*zSSIIb* copy number ratio (Table 5). The certified values were determined via collaborative characterization, which was performed by eight laboratories using the DBN9936/*zSSIIb* duplex ddPCR assay. ddPCR is a higher-order reference measurement method that offers copies of a particular nucleic acid sequence with unit one, which can be considered as traceable to the International System of Units (SI) [28,29,30]. In addition, the DBN9936/*zSSIIb* duplex ddPCR method had been sufficiently optimized and validated before the current characterization was performed [15]. Therefore, the certified values fulfill the highest standards of accuracy.

The uncertainties associated with the certified values were estimated in accordance with ISO 17034:2016 and ISO Guide 35:2017 [14,24]. The assigned uncertainty of the certified values consists of the uncertainties related to the characterization (*u*_char_), potential between-unit inhomogeneity (*u*_bb_), potential instability during dispatch (*u*_sts_), and long-term storage (*u*_lts_). These different contributions were combined to estimate the expanded uncertainties of the certified values using Equation (6). The certified values, together with their uncertainties, for the three property values of the three gDNA CRMs are shown in Table 5.

## 4. Discussion

China will soon begin promoting the commercial planting of GM crops, and it is likely that the accidental contamination of non-GMO products with GMO ingredients will increase. Hence, there is an urgent need to implement a labeling policy that specifies defined thresholds for GM content to protect the public’s right to know and to reduce the labeling costs incurred by producers. Agricultural products, food, and feed that contain GM ingredients at levels below the defined thresholds can be exempt from labeling.

The development and certification of RMs for GMOs are fundamental to the quantification of GMO content and the enforcement of a labeling policy. As a result, many CRMs for GMO quantification have been developed and certified to meet the labeling requirements set for products containing GMOs. Previously developed gDNA CRMs are mainly pure CRMs prepared from GM homozygous plants, such as GM rice Kefeng 6 and KMD [10,11]. Such gDNA CRMs are assigned two certified values, for copy number concentration and copy number ratio, to meet the calibration requirements for the measurement of GMO content. However, in the quantification of GMOs, RMs with defined GMO content equivalent to the threshold stipulated in the labeling legislation are also required as quality controls to monitor performance bias that may be present in the quantification process [4,13]. Hence, to meet the calibration and bias control requirements set for qPCR assays, we have developed not only a pure gDNA CRM (DBN9936a) but also two low-level gDNA CRMs (DBN9936b and DBN9936c) with copy number ratios of 3.5% and 1.2%, respectively, by quantitatively mixing gDNA from GM and non-GM plants.

CRMs with metrological traceability are essential for analytical traceability and the comparability of analytical results [31]. The validated DBN9936/*zSSIIb* duplex ddPCR assay was used for the homogeneity assessment, stability assessment, and collaborative characterization of this batch of gDNA CRMs. As an enumeration-based measurement procedure, the dPCR forms the basis of this primary reference measurement procedure for measuring DNA copy number concentration [3]. It has been widely used to characterize nucleic acid RMs [29,30] and reports the quantity of DNA with unit one in each reaction, where unit one is by nature an element of any system of units [3]. The main uncertainty contributions were considered and combined to estimate the expanded uncertainty associated with each certified value (*U*_CRM_) with a coverage factor k value of 2.0 in compliance with ISO 17034:2016. The certified values of this batch of CRMs can be considered traceable to unit one of the SI [32].

The duplex ddPCR technique, which synchronously detects transgenes and reference genes in the same reaction, eliminates the pipetting error between transgene and reference gene assays and has been proven to have higher accuracy than the simplex ddPCR technique [33]. The property values of the low-level CRM DBN9936c were accurately measured with satisfactory precision by duplex ddPCR during the assessment and characterization phases of this study. The measurement results demonstrated that the ddPCR can be used to perform reliable and accurate characterization of low-level CRM that has a GMO content as low as approximately 1%. The measurement of the DBN9936/*zSSIIb* copy number ratio by duplex ddPCR was not affected by the droplet volume (*v*) and dilution factor of the template (*d*) [3]; therefore, the uncertainty related to the characterization was reduced.

This batch of gDNA CRMs for use in DBN9936 event detection consisted of DBN9936a, DBN9936b, and DBN9936c, and three property values of each CRM were evaluated: DBN9936 copy number concentration, *zSSIIb* copy number concentration, and DBN9936/*zSSIIb* copy number ratio. The certified values, together with their expanded uncertainties, are summarized in Table 5. Therefore, in this study, a set of user-friendly gDNA CRMs has been developed and granted a certificate (GBW10266, GBW10267, GBW10268), and the strategy for producing this batch of gDNA CRMs can be used for production of other GMOs’ gDNA CRM. This batch of gDNA CRMs is available for other laboratories and can be directly used for calibration and quality control in qPCR assays to effectively ensure the accuracy, reliability, and comparability of analytical data over time and space.

## Figures and Tables

**Figure 1 foods-13-00747-f001:**
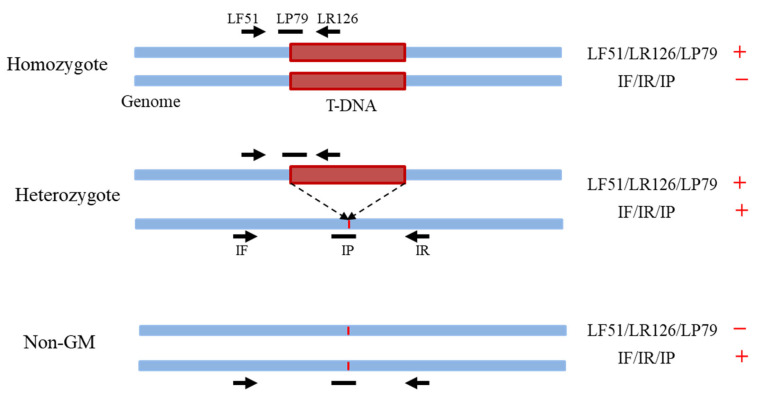
Diagram of identification of zygosity in GM materials. The blue rectangular box represents genomic DNA, the red rectangular box represents inserted T-DNA, the black arrow identifies the primer position, the black horizontal bar represents the probe position, the red vertical bar represents the insertion site of T-DNA, the plus sign represents positive result of PCR, and the minus sign represents negative result of PCR.

**Table 1 foods-13-00747-t001:** Results of the homogeneity testing of DBN9936a, DBN9936b, and DBN9936c.

Parameter	DBN9936a			DBN9936b			DBN9936c		
	DBN9936 (Copies/μL)	*zSSIIb* (Copies/μL)	Ratio	DBN9936 (Copies/μL)	*zSSIIb* (Copies/μL)	Ratio	DBN9936 (Copies/μL)	*zSSIIb* (Copies/μL)	Ratio
			(%)			(%)			(%)
Mean	34,293	34,804	98.53	2717	460,847	3.34	383	763,689	1.09
*MS* _between_	374,247	185,269	1.32	2717	460,847	0.019	383	763,689	0.004
*MS* _within_	259,031	201,751	1.24	1489	241,022	0.014	252	700,320	0.002
*F*	1.44	0.91	1.06	1.82	1.91	1.33	1.52	1.09	1.88
*F* _(0.05,14,30)_	2.04			2.04			2.04		
*s* _bb_	195.97	<0	0.16	20.23	270.69	0.04	6.61	145.34	0.03
*s* _wb_	508.95	449.17	1.11	38.59	490.94	0.12	15.87	836.85	0.04
*u* _bb*_	149.311	131.772	0.326	11.321	144.027	0.035	4.659	245.508	0.013
*u* _bb_	195.973	131.772	0.326	20.232	270.694	0.041	6.608	245.508	0.026
*u* _bb, rel_	0.0057	0.0038	0.0033	0.0074	0.0006	0.0122	0.0173	0.0003	0.0237

**Table 2 foods-13-00747-t002:** Results of the short-term stability study and the assessment of the related uncertainties.

RM	Storage Time (Days)	4 °C			25 °C			37 °C		
		DBN9936 (Copies/μL)	*zSSIIb* (Copies/μL)	Ratio (%)	DBN9936 (Copies/μL)	*zSSIIb* (Copies/μL)	Ratio (%)	DBN9936 (Copies/μL)	*zSSIIb* (Copies/μL)	Ratio (%)
DBN9936a	0	33,087	33,353	99.21	34,900	34,973	99.79	34,860	35,087	99.36
	3	35,207	35,547	99.05	33,980	33,927	100.16	34,047	34,153	99.69
	7	35,073	35,413	99.05	34,880	35,007	99.64	34,947	35,320	98.95
	14	33,693	33,840	99.58	35,207	35,547	99.05	34,447	34,860	98.82
	Mean	34,265	34,538	99.22	34,742	34,863	99.66	34,575	34,855	99.21
	*β* _1_	4.364	−5.697	0.029	47.212	70.545	−0.066	−7.091	11.091	−0.052
	*S*(*β*_1_)	121.605	129.170	0.021	52.022	61.404	0.028	48.121	58.338	0.028
	*t* _0.95,n−2_	4.303	4.303	4.303	4.303	4.303	4.303	4.303	4.303	4.303
	*t*_0.95,n−2_. *S*(*β*_1_)	523.225	555.774	0.088	223.832	264.2	0.121	207.047	251.007	0.122
	Conclusion	|*β*_1_| < *t*_0.95,n−2_. *S*(*β*_1_), stable								
	*u* _sts_	-	-	-	-	-	-	673.693	816.729	0.396
	*u* _sts,rel_	-	-	-	-	-	-	0.019	0.023	0.004
DBN9936b	0	1009	31,167	3.24	1039	32,253	3.22	1014	31,673	3.2
	3	1045	32,100	3.26	1002	31,340	3.2	1031	31,993	3.22
	7	1014	31,173	3.25	1003	30,907	3.25	1039	31,733	3.28
	14	1009	31,373	3.22	1017	31,027	3.28	1025	31,453	3.26
	Mean	1019	31,453	3.24	1015	31,382	3.24	1027	31,713	3.24
	*β* _1_	−0.988	−10.364	−0.002	−0.879	−76.545	0.005	0.6	−24.182	0.004
	*S*(*β*_1_)	1.9198	51.0499	0.0017	1.9088	46.1557	0.0019	1.1625	19.5025	0.0026
	*t* _0.95,n−2_	4.303	4.303	4.303	4.303	4.303	4.303	4.303	4.303	4.303
	*t*_0.95,n−2._ *S*(*β*_1_)	8.26	219.65	0.008	8.213	198.592	0.008	5.002	83.912	0.011
	Conclusion	|*β*_1_| < *t*_0.95,n−2_. *S*(*β*_1_), stable								
	*u* _sts_	-	-	-	-	-	-	16.275	273.035	0.036
	*u* _sts,rel_	-	-	-	-	-	-	0.016	0.009	0.011
DBN9936c	0	363	32,447	1.12	357	31,240	1.14	341	30,847	1.11
	3	345	31,093	1.11	366	31,393	1.17	333	29,967	1.12
	7	355	31,587	1.12	360	32,120	1.12	351	30,920	1.06
	14	344	32,493	1.06	370	32,367	1.14	347	32,373	1.06
	Mean	352	31,905	1.1	363	31,780	1.14	343	31,027	1.09
	*β* _1_	−0.939	32.485	−0.004	0.709	85.758	−0.001	0.758	135.697	−0.004
	*S*(*β*_1_)	0.799	76.384	0.002	0.444	20.439	0.002	0.751	65.899	0.002
	*t* _0.95,n−2_	4.303	4.303	4.303	4.303	4.303	4.303	4.303	4.303	4.303
	*t*_0.95,n−2_. *S*(*β*_1_)	3.440	328.654	0.009	1.910	87.944	0.009	3.230	283.540	0.010
	Conclusion	|*β*_1_| < *t*_0.95,n−2_. *S*(*β*_1_), stable								
	*u* _sts_	-	-	-	-	-	-	10.510	922.585	0.032
	*u* _sts,rel_	-	-	-	-	-	-	0.031	0.030	0.029

**Table 3 foods-13-00747-t003:** Results of the long-term stability study and the assessment of the related uncertainties.

Storage Time (Months)	DBN9936a			DBN9936b			DBN9936c		
	DBN9936 (Copies/μL)	*zSSIIb* (Copies/μL)	Ratio (%)	DBN9936 (Copies/μL)	*zSSIIb* (Copies/μL)	Ratio (%)	DBN9936 (Copies/μL)	*zSSIIb* (Copies/μL)	Ratio (%)
0	33,913	34,807	97.43	1089	32,080	3.39	333	31,267	1.07
1	33,593	34,340	97.83	1087	31,907	3.41	327	31,527	1.04
2	34,627	34,807	99.48	1107	32,093	3.45	335	30,680	1.16
4	34,807	35,007	99.43	1049	32,020	3.25	342	31,620	1.12
6	33,700	34,260	98.37	1087	32,120	3.38	337	31,347	1.07
12	34,653	34,833	99.48	1063	31,673	3.36	355	32,000	1.11
18	35,177	35,380	99.43	1064	32,087	3.32	310	29,800	1.04
Mean	34,424	34,776	98.95	1078	31,997	3.37	334	31,177	1.09
*β* _1_	33.906	13.704	0.034	−1.637	−5.16	−0.004	−0.525	−48.97	−0.002
*S*(*β*_1_)	38.441	25.237	0.06	1.137	10.578	0.004	0.905	44.222	0.003
*t* _0.95,n−2_	2.571	2.571	2.571	2.571	2.571	2.571	2.571	2.571	2.571
*t*_0.95,n−2_. *S*(*β*_1_)	98.815	64.874	0.154	2.924	27.191	0.01	2.326	113.677	0.008
Conclusion	|*β*_1_| < *t*_0.95,n−2_. *S*(*β*_1_), stable								
*u* _lts_	691.938	454.266	1.08	20.466	190.404	0.072	16.29	795.996	0.054
*u* _lts,rel_	0.020	0.013	0.011	0.019	0.006	0.021	0.049	0.026	0.050

**Table 4 foods-13-00747-t004:** Estimation of the uncertainty contributions in the characterization of the DBN9936 gDNA RMs.

RM	Property Value	Certified Value	*u* _A,rel_	*u* _λ,rel_	*u* _v,rel_	*u* _d,rel_	*u* _char,rel_
DBN9936a	DBN9936 (copies/μL)	33,801	0.022	0.011	0.015	0.006	0.029
	*zSSIIb* (copies/μL)	34,309	0.025	0.011	0.015	0.006	0.031
	DBN9936/*zSSIIb* (%)	98.42	0.014	0.015	-	-	0.020
DBN9936b	DBN9936 (copies/μL)	1050	0.032	0.036	0.015	0.006	0.051
	*zSSIIb* (copies/μL)	31,006	0.037	0.010	0.015	0.006	0.041
	DBN9936/*zSSIIb* (%)	3.39	0.032	0.038	-	-	0.050
DBN9936c	DBN9936 (copies/μL)	351	0.043	0.063	0.015	0.006	0.078
	*zSSIIb* (copies/μL)	31,059	0.027	0.010	0.015	0.006	0.033
	DBN9936/*zSSIIb* (%)	1.13	0.035	0.064	-	-	0.073

**Table 5 foods-13-00747-t005:** The certified values and their uncertainties for the three property values of the DBN9936 gDNA RMs.

RM	Property Value	Certified Value	*u* _char, rel_	*u* _bb,rel_	*u* _sts,rel_	*u* _lts,rel_	*u* _rel_	*u* _CRM_	*U* _CRM_
DBN9936a	DBN9936 (copies/μL)	3.38 × 10^4^	0.029	0.006	0.019	0.020	0.041	1387	0.28 × 10^4^
	*zSSIIb* (copies/μL)	3.43 × 10^4^	0.031	0.004	0.023	0.013	0.041	1419	0.29 × 10^4^
	DBN9936/*zSSIIb* (%)	98.4	0.020	0.003	0.004	0.011	0.024	2.32	4.7
DBN9936b	DBN9936 (copies/μL)	1.05 × 10^3^	0.051	0.019	0.016	0.019	0.060	63	0.13 × 10^3^
	*zSSIIb* (copies/μL)	3.10 × 10^4^	0.041	0.008	0.009	0.006	0.044	1349	0.27 × 10^4^
	DBN9936/*zSSIIb* (%)	3.39	0.050	0.012	0.011	0.021	0.057	0.19	0.39
DBN9936c	DBN9936 (copies/μL)	3.51 × 10^2^	0.078	0.019	0.031	0.049	0.099	35	0.70 × 10^2^
	*zSSIIb* (copies/μL)	3.11 × 10^4^	0.033	0.008	0.030	0.026	0.052	1606	0.33 × 10^4^
	DBN9936/*zSSIIb* (%)	1.13	0.073	0.024	0.030	0.050	0.096	0.11	0.22

## Data Availability

The original contributions presented in the study are included in the article/Appendix A, further inquiries can be directed to the corresponding author.

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
