# Peer review of "The Development of a Series of Genomic DNA Reference Materials with Specific Copy Number Ratios for The Detection of Genetically Modified Maize DBN9936"

_foods, 2024, doi:10.3390/foods13050747_

Round 1

Reviewer 1 Report

Comments and Suggestions for Authors

The authors have provided relevant information about development of reference materials for DBN9936 maize GMO event.  

I suggest the authors to elaborate on some points for clarification:

Lines 57-59. AOCS reference materials are not developed by AOCS. GMO developers provide RMs to AOCS. Correct the sentence to reflect that.

Line 128.  Change ‘The extracted gDNA was solved in 0.1×TE’ to ‘The extracted gDNA was dissolved in 0.1×TE’

Lines 133-134. Provide brief description of ddPCR instrument used (e.g., manufacturer, conditions used for PCR, etc.) for clarity.

Line 382-385. “The measurement data from the eight laboratories displayed a normal distribution. The arithmetic measurement data from the eight laboratories were assigned as the certified values of the three gDNA RMs”. I suggest presenting summary data of the certified values obtained by eight laboratories (supplementary information) – this is an important information as the certified values were obtained based on the data obtained from the eight laboratories.

Table S4. Indicate what 1, 2, 3, 4 and 5 freeze-thaw cycles are at the end of the Table for clarity.

Are the developed reference materials available for other labs to use (as sale or collaboration)? Please provide information in conclusion or where relevant.

Comments on the Quality of English Language

Minor editing needed

Author Response

The authors have provided relevant information about development of reference materials for DBN9936 maize GMO event. 

I suggest the authors to elaborate on some points for clarification:

Lines 57-59. AOCS reference materials are not developed by AOCS. GMO developers provide RMs to AOCS. Correct the sentence to reflect that.

Answer:We have correct this sentence according to the above suggestion. (line 62)

Line 128.  Change ‘The extracted gDNA was solved in 0.1×TE’ to ‘The extracted gDNA was dissolved in 0.1×TE’

Answer: The word “solved” was changed to “dissolved”.

Lines 133-134. Provide brief description of ddPCR instrument used (e.g., manufacturer, conditions used for PCR, etc.) for clarity.

Answer: We added the brief description of the duplex ddPCR ((e.g., manufacturer, conditions used for PCR, etc.). (line 158-162)

Line 382-385. “The measurement data from the eight laboratories displayed a normal distribution. The arithmetic measurement data from the eight laboratories were assigned as the certified values of the three gDNA RMs”. I suggest presenting summary data of the certified values obtained by eight laboratories (supplementary information) – this is an important information as the certified values were obtained based on the data obtained from the eight laboratories.

Answer: We add the data of collaborate characterization of eight laboratories in supplementary file 9 (Table S5).

Table S4. Indicate what 1, 2, 3, 4 and 5 freeze-thaw cycles are at the end of the Table for clarity.

Answer: According to the suggestion, we explained a freeze-thaw cycle at the end of the Table S4.

Are the developed reference materials available for other labs to use (as sale or collaboration)? Please provide information in conclusion or where relevant.

Answer: this batch of gDNA RMs has been granted a certificate, and can be available for other labs, which has been added to the last paragragh of this paper. (line 503-506)

Reviewer 2 Report

Comments and Suggestions for Authors

The manuscript presents original study. Experiments are well designed. Sufficient number of results are presented and the conclusions correspond to the results obtained. The study has great practical value as  the availability of certified reference materials is crucial for the monitoring of GMO products on the market and especially when a thresholds are applied. 

The manuscript presents original study on the development and certification of new reference material for novel maize GM event DBN9936. This CRMis aimed to be used for calibration and quality control for detection/quantification of DBN9936 for GMO montioring purposes. The study has great practical value as the availability of certified reference materials is crucial for the monitoring of GMO products on the market when legislation threshold is applied.

Introduction

The authors report preparation and certifiation of 3 different CRM with certified values 100%, 3,5% and 1,2%. It will be interesting to know why these certified values are selected. In GMO testing if thresholds are applied it is advantage to have CRM close to the legislation threshold value. It would be benefitial to comment what is the advantage of these certified GM levels in

this context.

Material and methods

The experiments are properly designed. The material and methods are described in sufficient details. However, the ddPCR method used is not described as the authors refer to previous publication. I recommend to add in the supplementary materials some details on the ddPCR conditions – mastermix, PCR programme, instruments. The authors have also performed relevant statistical analysis of the obtained data. I have one remark on the statistical analysis: the software tool used to perform the statistical testse;g. ANOVA is not mentioned in the manuscript, so it should be added.

Results

Sufficient number of results are presented. The results are summarized in tables. I recommend to improve the format of tables 4 and 5, in order to separate better the data for each reference material. The current format is difficult to read.

Discussion

The conclusions are supported by the presented results. However, it would be benefitial if the authors can briefly comment on the future prospectives for production of this kind of materials as they say in the introduction (raws 88-90) that through the production of these CRM they aim at establisshing a platform for production of series of GMO gDNA CRM.

Author Response

The manuscript presents original study. Experiments are well designed. Sufficient number of results are presented and the conclusions correspond to the results obtained. The study has great practical value as the availability of certified reference materials is crucial for the monitoring of GMO products on the market and especially when a thresholds are applied.

Answer: Thanks for the reviewer’s approval of this manuscript.

Introduction

The authors report preparation and certifiation of 3 different CRM with certified values 100%, 3,5% and 1,2%. It will be interesting to know why these certified values are selected. In GMO testing if thresholds are applied it is advantage to have CRM close to the legislation threshold value. It would be benefitial to comment what is the advantage of these certified GM levels in this context.

Answer: Thanks for the suggestion. In this study, 3 CRMs with certified values 100%, 3.39% and 1.13% were developed, the 100% CRM is used for calibration, the 3.39 % is close to the threshold value of China, and the 1.13% is close to that of EU. The two low level CRMs are used for bias control in quantification. In Introduction section we emphasize to “develop low-level gDNA CRMs equivalent to threshold values” (line 80). In Preparation of the gDNA RMs (2.4.), the sentence “approximately equivalent to the threshold value of China and EU”, was added (line 148).

Material and methods

The experiments are properly designed. The material and methods are described in sufficient details. However, the ddPCR method used is not described as the authors refer to previous publication. I recommend to add in the supplementary materials some details on the ddPCR conditions – mastermix, PCR programme, instruments. The authors have also performed relevant statistical analysis of the obtained data. I have one remark on the statistical analysis: the software tool used to perform the statistical testse;g. ANOVA is not mentioned in the manuscript, so it should be added.

Answer: The details on the ddPCR conditions – mastermix, PCR programme, instruments were added to the material and method section (line 158-162). The software tool for statistical analysis was provided (line 163).

Results

Sufficient number of results are presented. The results are summarized in tables. I recommend to improve the format of tables 4 and 5, in order to separate better the data for each reference material. The current format is difficult to read.

Answer: We changed the dashed lines between different RMs into solid lines in Tables 4 and 5.

Discussion

The conclusions are supported by the presented results. However, it would be benefitial if the authors can briefly comment on the future prospectives for production of this kind of materials as they say in the introduction (raws 88-90) that through the production of these CRM they aim at establisshing a platform for production of series of GMO gDNA CRM.

Answer: The developed gDNA RMs in this study was granted a certificate by National CRM Management Committee, demonstrating the strategy for producing this batch of CRMs is applicable. We provided the following content “a set of user-friendly gDNA CRMs has been developed and approved a certificate (GBW10266, GBW10267, GBW10268), the strategy for producing this batch of gDNA CRMs can be used for production of other GMOs’ gDNA CRM” in conclusion section. (line 503-505)